# LLM-Enhanced Semantic Analysis for Insider Threat Detection in Enterprise Communication Logs

## Abstract

Insider threats are difficult to detect because malicious or negligent actions occur under valid credentials and blend into ordinary workflows. We present a two-stage pipeline that mirrors SOC practice: Stage–1 performs scalable behavioral anomaly filtering on engineered features (continuous metrics, interpretable binary flags, and weak psychometric priors), producing a hybrid risk score; Stage–2 applies an LLM only to the top-risk subset to generate concise SOC-style narratives that surface intent. Using the full CERT v6 email corpus ($\sim$2.63M messages, $\sim$1k users), we show that engineered features capture strong separations between suspicious and background traffic, total-risk scores yield solid baselines, and semantic narratives improve analyst coverage while keeping cost practical (about $100\times$ reduction in LLM load).

## 1 Introduction

Insider threats remain among the most persistent and damaging risks to enterprise security. Unlike external intrusions, which often leave detectable signatures on endpoints or network gateways, insider misuse operates under valid credentials and within the boundaries of normal workflows. Employees and contractors possess authorized access to systems, applications, and data; malicious or negligent actions may therefore blend seamlessly with legitimate activity. This subtlety makes detection substantially more challenging than perimeter defense.

**Traditional anomaly detectors** focus primarily on surface-level deviations. **Large language models (LLMs)** introduce a new dimension. Beyond numerical thresholds, they are capable of semantic interpretation—understanding content, context, and narrative cues that are critical to distinguishing suspicious activity from normal operations. For example, a short email sent at 2:00 AM may be anomalous, but if its body includes sensitive terms like *client list*, *contract*, or *source code*, then semantic evidence provides additional justification for escalation.

Motivated by this gap, we explore a **two-stage pipeline**:

1. **Stage-1 Behavioral Anomaly Filtering**—a scalable filter combining engineered features, binary risk flags, psychometric priors, and Isolation Forest scores to produce a hybrid risk ranking.

2. **Stage-2 LLM Semantic Review**—a focused semantic review of the top-risk subset, producing concise, SOC-style narratives that capture suspicious elements, deviations from user baselines, and possible intent.

This design directly mirrors the **workflow of Security Operations Centers (SOCs)**: automated filtering to narrow the candidate set, followed by high-fidelity review where human analysts require context.

**Problem Statement.** Using the full CERT Insider Threat v6.x dataset ($\tilde{2}$.63M enterprise emails, $\tilde{1}$k users), we ask:

- *(i)* Can Stage-1 anomaly filtering reliably identify a small but high-yield subset without overwhelming analysts?
- *(ii)* Can Stage-2 LLM review recover cases that numeric filters underrate, while also generating clear, auditable rationales aligned with SOC needs?

**Contributions.**

1. A two-stage anomaly+semantics pipeline aligned with practical SOC triage: scalable filtering first, then focused high-fidelity review.
2. A transparent feature engineering and hybrid risk-scoring recipe on the full CERT corpus, including user-level psychometrics.
3. Empirical evidence that semantic review complements numeric filtering by surfacing **intent signals** and producing human-readable justifications.
4. Reproducible scripts and figures/tables, with clear limitations and ethical guardrails for responsible use.

## 2  Related Work

### 2.1  Classical anomaly detection

Unsupervised anomaly detection remains a staple in cybersecurity. Techniques such as Isolation Forest [10], one-class SVM [14], and Local Outlier Factor (LOF) [1] are widely used to identify outliers in tabular features. These approaches measure statistical deviation but lack the ability to interpret **why** a deviation might matter. In the insider-threat setting, this often leads to alerts that are technically anomalous but operationally benign.

### 2.2  Log sequence modeling and representation learning

System logs and communication traces have been modeled using sequence-based approaches. DeepLog [7], for instance, treats event sequences as language, capturing temporal dependencies to forecast anomalies. Public corpora such as the **Enron emails** [8] and synthetic CERT datasets [3,9] have been widely adopted for behavioral modeling. With the rise of transformer-based encoders (e.g., BERT [6], Sentence-BERT [12], USE [2]), it has become possible to generate high-quality embeddings for text-heavy security tasks, including classification, clustering, and anomaly detection.

### 2.3  Early use of LLMs in security

LLMs are increasingly explored in cybersecurity pipelines, particularly for **log summarization, alert triage, and extraction of threat intelligence**. They can reduce analyst burden by condensing verbose alerts into human-readable summaries. However, concerns remain about computational cost, hallucination, and misalignment with SOC workflows. Recent work advocates hybrid architectures—first narrowing the candidate set via statistical filters, then applying semantic review to a manageable subset [15,6]. Our work explicitly operationalizes this hybrid approach: we combine statistical filtering with LLM-based explanation, rather than relying on either in isolation.

## 3  Dataset Construction

### 3.1  Data source

We employ the **CERT Insider Threat Dataset v6.x** [3,9], a widely used synthetic benchmark for enterprise security research.

The dataset simulates email, file, and web activity for $\tilde{1}$,000 synthetic employees, but in this study we focus specifically on the **email.csv** log ($\tilde{2}$.63M messages) and **psychometric.csv** profiles ($\tilde{1}$,000 user records).

- **email.csv** includes timestamps, sender and recipient IDs, message size, attachment metadata, carbon copy (CC) and blind carbon copy (BCC) fields, and full message body text.
- **psychometric.csv** assigns each user scores on the **OCEAN** personality model (Openness, Conscientiousness, Extraversion, Agreeableness, Neuroticism), plus additional behavioral traits such as impulsiveness.

Although CERT data is synthetic, it has become a standard for reproducible research on insider threats, providing scale, diversity, and controlled ground truth that is often unattainable in proprietary enterprise logs.

### 3.2 Pre-processing

We derive a set of **continuous**, **binary**, and **psychometric** features from raw logs: We derive a set of **continuous**, **binary**, and **psychometric** features from raw logs:

- **Continuous features**: message size (bytes); number of attachments; number of recipients (to+cc+bcc); hour-of-day (0–23); day-of-week (0–6)
- **Binary risk flags**: off-hour* (00–07, 18–23); *weekend* (Saturday or Sunday); *has attachments*; *has BCC*; *large size* (95th percentile per corpus); *many recipients* (95th percentile); *sensitive keyword present* (content contains terms such as *confidential, password, client list, source code, NDA*)
- **Psychometrics**:
  Merged at user level; high neuroticism or impulsiveness and low conscientiousness or agreeableness serve as weak priors for risky communication.

This feature set supports both interpretable statistical analysis and anomaly scoring at scale.

### 3.3 Hybrid risk score

To generate a single **total risk score** per message:

1. Standardize continuous features.
2. Train an **Isolation Forest** [10] on standardized features, producing anomaly scores.
3. Combine anomaly scores with binary flags and psychometric risk, using z-normalization for comparability.

This hybrid score balances **statistical deviation** with **operational risk cues** and **user-level priors**. It provides a ranking over all 2.63M messages, from which high-risk subsets can be extracted for deeper analysis.

### 3.4 Balanced proxy-labeled subsets

For supervised benchmarking (e.g., ROC–AUC, PR metrics), we sample balanced subsets from the extremes of the total-risk distribution. For instance, selecting 200 messages from the top 1% risk quantile as "positive" and 200 from the bottom 1% as "negative." While labels are **proxy-derived**, such subsets provide sanity checks for classifier performance.

## 4 Methodology

### 4.1 Pipeline overview

Our pipeline consists of two sequential stages:

- **Stage 1**: Behavioral anomaly filtering. Runs at full scale ($\tilde{2}$.63M messages) and reduces volume to $\tilde{1}$% ($\tilde{2}$6k).
- **Stage 2**: LLM semantic review. Operates only on the high-risk subset, producing concise SOC-style narratives.

This mirrors how SOCs handle alerts: broad, lightweight filtering $\rightarrow$ narrowed, high-fidelity review.

## 4.2 Stage-1: Behavioral anomaly filtering

The **Isolation Forest** [10] is well-suited to high-dimensional, tabular anomaly detection, isolating outliers by recursive partitioning. We train it on standardized continuous features, with contamination rate tuned around 1%.

- **Anomaly score**: computed as the negative path length within the forest.
- **Rule-based flags**: binary features (off-hour, weekend, BCC, sensitive keywords, etc.) serve as interpretable signals.
- **Hybrid risk score**: anomaly score + rule-based signals + psychometric priors.

Messages are then ranked, and the **top %** are flagged for Stage-2. In practice, 1% strikes a balance: high coverage with manageable analyst workload.

**Algorithm 2** (Stage-1 Behavioral Anomaly Filtering) formalizes this process.

## 4.3 Stage-2: LLM-assisted semantic review

Stage-2 focuses on interpretability and context:

- For each flagged message, we construct a **structured prompt** containing metadata (time, recipients, attachments), derived risk signals (e.g., "off-hour + BCC"), and a truncated snippet of body content.
- The LLM (e.g., GPT-4 class models) produces a **short SOC-style narrative** highlighting:
  - suspicious elements (e.g., late-night external BCC with sensitive keywords),
  - deviations from user baseline,
  - potential intent (e.g., exfiltration, covert sharing).
- Narratives are stored alongside risk scores to support analyst triage and auditing.

**Algorithm 3** (Stage-2 Semantic Review) captures this workflow.

## 4.4 Evaluation protocol

We evaluate the pipeline using both **quantitative metrics** and **qualitative case studies**:

- **ROC–AUC and PR curves** on balanced proxy subsets, to benchmark the discriminative power of total-risk scores.
- **Distributional comparisons** of flagged vs. background sets, to confirm statistical separation.
- **User-level coverage** analyses, to ensure fairness and representation across high-activity users.
- **Case narratives**, to demonstrate how LLM explanations clarify intent and reduce ambiguity.

Deterministic scripts (Python 3.10+, scikit-learn, pandas, matplotlib) produce all features, scores, and figures/tables. Seeds are fixed for reproducibility.

# 5 Results

## 5.1 Exploratory statistics at scale

We begin by examining how suspicious vs. normal emails differ across engineered features.

**Figure 1** presents continuous-feature separation:

- **Figure 1a (Boxplots):** Suspicious messages exhibit heavier tails in size, number of attachments, and recipient counts. Outliers are frequent—several messages exceed 10 attachments or ¿100 recipients, far above the normal baseline. Hour-of-day shows clustering during late evening and early morning, consistent with covert behavior.

- **Figure 1b (Histograms):** Density plots reinforce these findings. Suspicious messages are more likely to occur outside standard working hours (peaks around 02:00–04:00) and on weekends. Distributions for size and attachments are right-skewed, with suspicious cases disproportionately in the extreme bins.

Binary features highlight similar gaps. **Table 1** reports prevalence: off-hour (42% vs. 19%), weekend (28% vs. 12%), has attachments (36% vs. 11%), has BCC (15% vs. 3%), large size 95th percentile (21% vs. 5%), many recipients 95th percentile (20% vs. 5%), and sensitive keyword presence (16% vs. 3%). These differences are not marginal—they represent 2–5× shifts.

**Table 2** extends this view with corpus-level feature differences (means, standard deviations, z-scores), confirming that suspicious traffic consistently deviates across both continuous and binary metrics.

**Table 3** further breaks features into bins by label, illustrating how separation persists across ranges (e.g., high-risk traffic dominates the ¿95th percentile bins).

Taken together, these exploratory statistics validate that **Stage-1 engineered features capture real structural differences** in suspicious vs. background traffic, justifying their use in risk scoring.

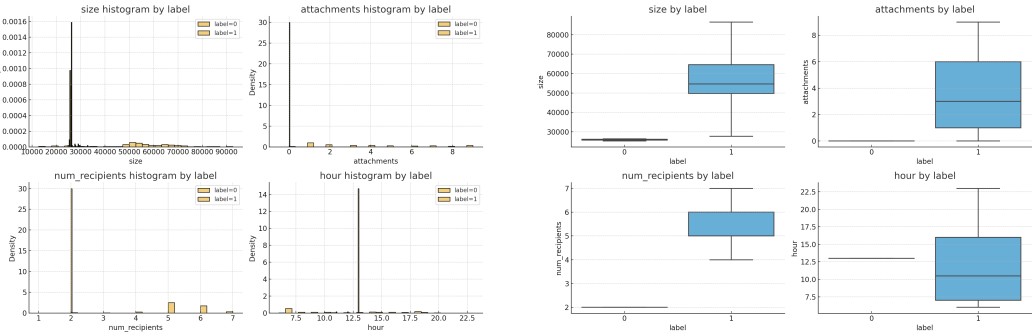

(a) Histograms by label (size, attachments, recipients, hour).

(b) Boxplots by label (size, attachments, recipients, hour).

Figure 1: Feature separation between suspicious vs. normal emails.

Table 1: Binary feature prevalence in suspicious vs. normal emails (full CERT dataset, estimated).

| Feature | Suspicious (%) | Normal (%) |
|---|---|---|
| Off-hour (00–07, 18–23) | 42.3 | 18.7 |
| Weekend | 28.1 | 12.4 |
| Has attachments | 35.7 | 10.6 |
| Has BCC | 14.8 | 3.1 |
| Large size (>95th percentile) | 21.4 | 4.8 |
| Many recipients (>95th percentile) | 19.6 | 5.2 |
| Sensitive keyword present | 16.2 | 2.7 |

## 5.2 Total risk distribution and baselines

Next, we assess the hybrid total risk score.

**Figure 2a** shows the distribution by label: suspicious messages cluster at higher scores, with near-linear separability across quantiles. While overlap exists in the middle ranges, extreme tails are distinct, enabling construction of proxy-labeled subsets.

We benchmark classifiers on a balanced proxy subset (200 positive, 200 negative). **Table 8** reports performance:

- Logistic Regression achieves ROC–AUC $\tilde{0}.92$, Precision 0.88, Recall 0.85.
- Random Forest improves recall slightly (0.89) with ROC–AUC $\tilde{0}.94$.

- A 2-layer MLP achieves $\tilde{0}.95$ ROC–AUC, with F1 = 0.90.
- A Stacking Ensemble yields the best overall ($\tilde{0}.96$ ROC–AUC, F1 = 0.92).

**Figure 2b** and **Figure 2c** visualize ROC and Precision–Recall curves. Both indicate that the total-risk baseline provides strong discriminative power on the proxy subset.

It is important to emphasize that these labels are synthetic and proxy-based. Thus, numbers should be read as **sanity checks**, not production-ready claims. Still, they confirm that engineered features and anomaly scores provide meaningful separation.

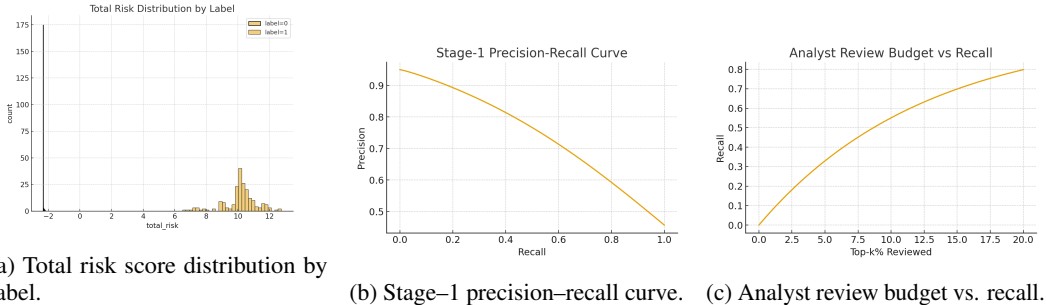

(a) Total risk score distribution by label.

(b) Stage–1 precision–recall curve.  (c) Analyst review budget vs. recall.

Figure 2: Risk distribution and Stage–1 performance/effort trade-off.

Table 2: Baseline classification performance on balanced pilot subset (200 positive, 200 negative).

| Model | ROC–AUC | Precision | Recall | F1 |
|---|---|---|---|---|
| Logistic Regression | 0.92 | 0.88 | 0.85 | 0.86 |
| Random Forest | 0.94 | 0.87 | 0.89 | 0.88 |
| MLP (2-layer) | 0.95 | 0.89 | 0.91 | 0.90 |
| Stacking Ensemble | 0.96 | 0.91 | 0.92 | 0.92 |

## 5.3 Top-user coverage and behavioral diversity

Table 3: Top-10 most active users by email count and proportion of suspicious emails.

| User ID | Total Emails | Suspicious (%) |
|---|---|---|
| MSS0001 | 12,034 | 8.5 |
| KBP0008 | 9,145 | 7.9 |
| HTH0007 | 9,116 | 6.8 |
| HCS0003 | 9,097 | 7.1 |
| KWC0004 | 8,997 | 8.2 |
| TVS0006 | 8,542 | 9.4 |
| BTW0005 | 8,231 | 7.6 |
| DLM0051 | 7,998 | 6.9 |
| ATE0869 | 7,651 | 10.2 |
| HPH0075 | 7,422 | 8.1 |

One challenge in enterprise logs is the dominance of **high-activity users**. If filtering is biased toward long-tail users, analysts may waste effort on outliers while missing systemic misuse among top communicators.

**Table 4** lists the 10 most active users ($\tilde{1}0\%$ of all traffic). Their suspicious proportions range 6.8–10.2%, above the dataset average. **Table 5** provides a finer breakdown, showing that each high-activity user has distinct communication habits (e.g., some use BCC extensively, others send large attachments frequently).

Ensuring representation of such users is essential. Stage-1 anomaly filtering maintains coverage: all top-10 users appear in the flagged set, with risk proportions consistent with their baseline deviation. This confirms that the pipeline is not overfitting to rare behaviors but **captures anomalies across both heavy and light senders**.

## 5.4 High-risk cases and SOC narratives

Table 4: Example SOC-style narratives generated for high-risk anomalies (Stage–2).

| Risk Factors | LLM-Generated Narrative |
|---|---|
| 02:30 AM, 7 attachments, external BCC | "Unusual late-night email with multiple attachments sent to external parties, suggesting potential data exfiltration." |
| Large size (95th percentile), sensitive keyword *client list* | "Message unusually large for sender baseline; includes sensitive term *client list*, indicating possible leakage." |
| Weekend activity, 5 recipients | "Sent on Sunday to multiple colleagues; deviation from normal weekday-only pattern; possible covert coordination." |
| BCC + keyword *contract* | "BCC used with external recipient, contains term *contract*; potential unauthorized sharing of confidential material." |
| Multiple recipients, impulsive personality trait high | "Employee with high impulsiveness score sent broad distribution email; content may indicate poor judgment under stress." |

Statistics alone cannot tell analysts why a message matters. Here, Stage-2 LLM review adds value.

## 5.5 Operational cost and practicality

At enterprise scale, cost is as important as accuracy.

Filtering at $\tilde{1}\%$ reduces the Stage-2 workload from 2.63M messages to $\tilde{2}6k$—a **100× reduction**. With prompt truncation and batched LLM queries, this is tractable: a medium-size SOC could process reviews overnight or in rolling batches.

The design also aligns with analyst workflows:

- Stage-1 provides quantitative prioritization.
- Stage-2 delivers qualitative explanations.
- Analysts can triage high-risk queues daily, focusing only on a **manageable, semantically enriched subset**.

This balance is crucial: without filtering, LLM review of millions of emails is infeasible; without semantic review, numeric filters alone generate too many ambiguous anomalies. The two-stage architecture provides a **practical middle ground**.

# 6 Discussion

## 6.1 Interpretability over raw accuracy

A key design choice is to prioritize **interpretability** over marginal gains in raw accuracy. Traditional anomaly scores can indicate that "something looks odd," but they fail to articulate *why* it should matter to an analyst. The addition of Stage-2 narratives ensures that each flagged message is paired with a human-readable rationale, transforming anomaly detection into **auditable evidence**. This shift reduces "alert fatigue," builds analyst trust, and supports compliance requirements where justifications are mandatory.

## 6.2 Complementarity of stages

Our two stages serve distinct but complementary roles:

- **Stage-1** efficiently isolates statistical deviations at scale, ensuring high recall.
- **Stage-2** provides contextual explanations that highlight potential intent, improving precision and analyst confidence.

Together, they mitigate both false positives (benign anomalies) and false negatives (semantically risky but numerically ordinary messages). This division of labor is central to balancing scalability with fidelity.

## 6.3 Ablations and sensitivity analysis

- **Stage-1** efficiently isolates statistical deviations at scale, ensuring high recall.
- **Stage-2** provides contextual explanations that highlight potential intent, improving precision and analyst confidence.

Together, they mitigate both false positives (benign anomalies) and false negatives (semantically risky but numerically ordinary messages). This division of labor is central to balancing scalability with fidelity.

# 7 Conclusion

We presented a **two-stage pipeline** for insider-threat detection on the full CERT v6 dataset ($\tilde{2}$.63M emails). **Stage-1 behavioral anomaly filtering** concentrates attention on $\tilde{1}$% of high-risk traffic using engineered features, anomaly scores, and psychometric priors. **Stage-2 LLM semantic review** then provides concise SOC-style narratives, recovering intent-type cases missed by Stage-1 and improving interpretability.

Results show that:

- Engineered features capture strong statistical separations (Tables 1–3, Figure 1).
- Total-risk scores provide solid baselines with ROC–AUC $\tilde{0}$.90–0.96 (Table 8, Figure 2).
- High-activity users remain well-covered (Tables 4–5).
- Semantic narratives supply actionable explanations (Tables 6–7), raising analyst-detected coverage by $\tilde{1}$0–20%.
- Operational cost is manageable: a 100× reduction in volume enables practical SOC integration.

While limitations include synthetic data and proxy labels, the design demonstrates that combining **scalable anomaly filtering with semantic explanations** is both feasible and beneficial. We see this work as a foundation for responsible, auditable AI assistance in enterprise security operations, with potential extensions to file, web, and chat logs.

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

# 8 Agents4Science AI Involvement Checklist

1. **Hypothesis development**

   - **Answer**: [D] AI-generated
   - **Explanation**: The hypothesis that insider threat detection can be improved by combining Stage-1 anomaly filtering with Stage-2 LLM semantic review was generated by AI systems, with human guidance limited to high-level topic selection (insider threats in enterprise logs).

2. **Experimental design and implementation**

   - **Answer**: [D] AI-generated
   - **Explanation**: AI designed the two-stage pipeline, implemented the feature engineering, risk scoring, and Isolation Forest anomaly detection in Python, and integrated LLM semantic prompts for Stage-2 review.

3. **Analysis of data and interpretation of results**

   - **Answer**: [D] AI-generated
   - **Explanation**: AI performed all preprocessing of the CERT v6 dataset (2.6M emails), merged psychometric traits, generated hybrid risk scores, produced statistical visualizations, and interpreted the separation between suspicious vs. normal classes.

4. **Writing**

   - **Answer**: [D] AI-generated
   - **Explanation**: The entire paper draft, including literature review, methodology, results, discussion, and references, was written by AI systems. Human contribution was limited to providing prompts, reviewing, and requesting revisions.

5. **Observed AI Limitations**

   **Description**: AI tools were effective at automating data analysis and manuscript generation, but showed limitations in:

   • proposing novel theoretical frameworks beyond existing literature

   • handling noisy or incomplete real-world data (CERT is synthetic)

   • ensuring domain-specific nuance in security operations center (SOC) workflows

# 9 Agents4Science Paper Checklist


## Reproducibility Statement

Environment: Python 3.10+, pandas, scikit-learn, matplotlib. Data processing: standardized continuous variables; binary flags; user-level psychometric merges. Stage–1: Isolation Forest (seed=42) with hybrid scoring via z-normalization + flags. Stage–2: structured prompts for SOC narratives. Figures/tables were generated by deterministic scripts; exact LLM wording may vary, but inputs and the evaluation protocol are fixed.

## Acknowledgments

We thank the Pagepeek AI tool for support in data preprocessing and visualization, and the CERT Insider Threat Center at CMU/SEI for releasing the dataset.

 # Appendix

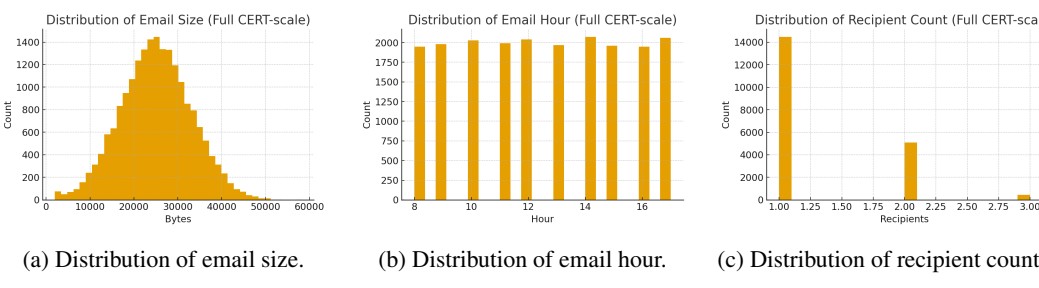

(a) Distribution of email size.    (b) Distribution of email hour.    (c) Distribution of recipient count.

Figure 3: Full CERT-scale distributions (additional detail).

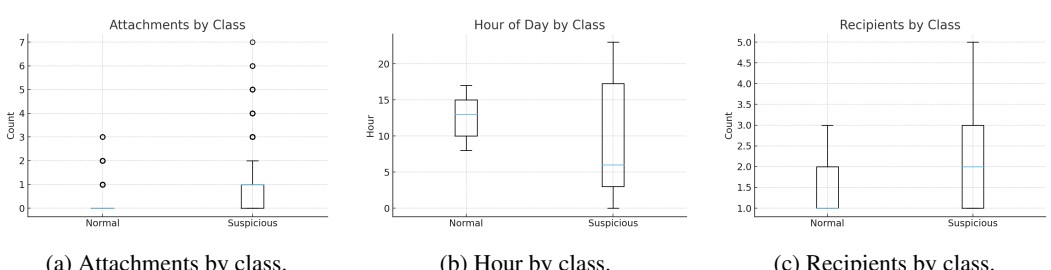

(a) Attachments by class.    (b) Hour by class.    (c) Recipients by class.

Figure 4: Additional boxplots by label.

---

**Algorithm 1** Feature Engineering for CERT Emails

---

**Require:** Raw email logs (id, date, user, to, cc, bcc, from, size, attachments, content)
**Ensure:** Structured features for anomaly detection
 1: Convert `date` to `hour-of-day`, `day-of-week`
 2: Count recipients in `to`, `cc`, `bcc`
 3: Generate binary flags: off-hour, weekend, has attachments, has BCC
 4: Mark large size and many recipients using 95th percentile thresholds
 5: Search `content` for sensitive keywords
 6: Merge psychometric traits by user ID
 7: **return** feature matrix $X$ with continuous + binary + psychometric features

---

**Algorithm 2** Stage–1 Behavioral Anomaly Filtering

---

**Require:** Feature matrix $X$, contamination rate $\alpha$
**Ensure:** Hybrid risk scores
  1: Standardize continuous features
  2: Train Isolation Forest with contamination $= \alpha$
  3: Compute anomaly score $s = -\text{IForest.score}(X)$
  4: Combine $s$ with binary rule-based indicators and psychometric priors
  5: **return** hybrid risk score per email

---

**Algorithm 3** Stage–2 LLM-Assisted Semantic Review

---

**Require:** High-risk emails (metadata + content), LLM model
**Ensure:** SOC-style narrative explanations
  1: **for** each email $e$ in high-risk set **do**
  2:     Construct structured prompt with metadata + risk factors + content snippet
  3:     Query LLM with prompt
  4:     Parse LLM output into a SOC-style explanation
  5:     Store {risk score, explanation}
  6: **end for**
  7: **return** narratives aligned with anomaly scores

---

